# Three in One: The Potential of *Brassica* By-Products against Economic Waste, Environmental Hazard, and Metabolic Disruption in Obesity

**DOI:** 10.3390/nu13124194

**Published:** 2021-11-23

**Authors:** José P. Castelão-Baptista, Ana Barros, Tânia Martins, Eduardo Rosa, Vilma A. Sardão

**Affiliations:** 1CNC—Center for Neuroscience and Cell Biology, CIBB—Centre for Innovative Biomedicine and Biotechnology, University of Coimbra, 3004-504 Coimbra, Portugal; castelao.baptista@gmail.com; 2Department of Life Sciences, Faculty of Sciences and Technology, University of Coimbra, 3000-456 Coimbra, Portugal; 3Centre for the Research and Technology of Agro-Environmental and Biological Sciences, University of Trás-os-Montes and Alto Douro, 5000-801 Vila Real, Portugal; abarros@utad.pt (A.B.); taniam@utad.pt (T.M.); erosa@utad.pt (E.R.); 4Inov4Agro—Institute for Innovation, Capacity Building and Sustainability of Agri-Food Production, University of Trás-os-Montes and Alto Douro, 5000-801 Vila Real, Portugal; 5Faculty of Sport Science and Physical Education, University of Coimbra, 3040-248 Coimbra, Portugal

**Keywords:** *Brassica* by-products, obesity, sulforaphane, oxidative stress

## Abstract

A large amount of waste is generated within the different steps of the food supply chain, representing a significant loss of natural resources, plant material, and economic value for producers and consumers. During harvesting and processing, many parts of edible plants are not sold for consumption and end up as massive waste, adding environmental hazards to the list of concerns regarding food wastage. Examples are *Brassica oleracea* var. *Italica* (broccoli) by-products, which represent 75% of the plant mass. A growing concern in the Western world is obesity, which results from incorrect lifestyles and comprises an extensive array of co-morbidities. Several studies have linked these co-morbidities to increased oxidative stress; thus, naturally occurring and readily available antioxidant compounds are an attractive way to mitigate metabolic diseases. The idea of by-products selected for their biomedical value is not novel. However, there is innovation underlying the use of *Brassica* by-products in the context of obesity. For this reason, *Brassica* by-products are prime candidates to be used in the treatment of obesity due to its bioactive compounds, such as sulforaphane, which possess antioxidant activity. Here, we review the economic and health potential of *Brassica* bioactive compounds in the context of obesity.

## 1. Introduction

In 1997, the World Health Organization (WHO) announced that obesity reached epidemic proportions worldwide [1]. Since then, obesity has risen at an alarming rate and is becoming a significant public health concern with vast social costs [2]. Obesity leads to the development of metabolic disorders, such as type 2 diabetes, non-alcoholic fatty liver disease, hypertension, coronary heart disease, several cancers, and psychological problems [3,4,5]. According to WHO, over 1.9 billion adults were considered overweight in 2016, and out of those, over 650 million were obese [6]. Unfortunately, obesity is still rising, with a 3-fold increase in incidence between 1975 and 2016 and spreading worldwide, making these numbers even more worrisome [6]. In 2014, the impact of obesity on the world economy was estimated to be around 2.0 trillion dollars (US). Thus, the development of new strategies to mitigate this global problem is imperative.

In addition to the negative impact that these economic and health issues have on society, one more factor negatively affects the lives of obese patients: the social stigma. Many of these patients are associated with negative misconceptions: obese people are lazy or weak-willed, unsuccessful, with low self-discipline and willpower, and purposely sabotage their weight-loss treatment [7]. These misconceptions give rise to discrimination against obese individuals in many aspects of their lives, including in health care facilities, educational institutions, workplace and job interviews, their representation in the media, and even in social situations [7]. Although some of these discriminatory attitudes may have a benevolent basis, as an incentive for the targeted person to become motivated to lose weight, the desired outcome may not be achieved, as there is much debate as to whether social stigma fails to change the subject’s behavior [8] or if social pressure will result in the modification of the patients’ lifestyle [9].

Most obesity cases result from an excess of energy consumption (dietary intake) relative to the expense of energy (energy loss via metabolic and physical activity). However, other underlying causes for this chronic disease result from complex interactions between genetic, behavioral, and environmental factors correlating economic and social status and lifestyles that characterize 21st-century living [10,11]. One of these complex conjunctions of factors is the constructed environment in developed countries that promotes overeating. Affordable, very tasty, conveniently prepared, and accessible in large quantities, highly caloric and fat-laden foods contribute to an elevated daily caloric intake [12], which is not offset by energy loss since, at the same time, physical activity levels have been decreasing over the past decades [13]. In developed countries, families stricken with economic issues tend to consume more of these highly caloric, easily preparable, and cheap meals due to their low income or lack of time due to demanding work schedules [14,15].

During the COVID-19 pandemic [16], lockdown measures forced gymnasiums, parks, playgrounds, and other places of physical activity practice to close down, meaning that many individuals could not participate in physical activities outside of their residences during an extended period. Furthermore, many individuals were less physically active under lockdown, had increased screen times, and adopted irregular sleep patterns and worse diets. The combination of these factors resulted in weight gain and loss of fitness. Once again, lower-income families were more affected by these stay-at-home rules due to the factors mentioned before and because these families usually live in sub-standard homes and more confined spaces, making it harder to practice physical exercise [17]. Since no studies have been conducted yet, the full scope of how COVID-19 affected obesity rates is unclear, but the outcome looks grim.

Food-related research that has recently raised considerable interest is the potential of natural products to counteract obesity [18]. Recommended therapeutic strategies in obese individuals include restricting simple carbohydrates and saturated fatty acid intake, increased physical activity, and the administration of anti-obesity drugs. Synthetic drugs have drawbacks in that they are relatively costly and their prolonged use may have adverse long-term side effects [18]. Therefore, alternative therapies using naturally occurring compounds have been proposed [19]. *Brassica* is a genus of plants in the *Cruciferae* family called Brassicaceae, consisting of about 350 genera and nearly 3500 species [20]. Several species that belong to the Brassicaceae family represent an essential part of the human diet worldwide. When regularly consumed, they have been found to exert health-promoting effects, like reducing the risk of chronic diseases, mainly cardiovascular diseases and several types of cancer [21,22]. These effects have been associated with the presence in these plants of phenolics, glucosinolates, carotenoids, tocopherols, and ascorbic acid, well-known compounds with antioxidant properties [23]. Moreover, in the last years, *in vivo* and *in vitro* studies have demonstrated the anti-obesity potential effect of isothiocyanates present in *Brassicas* [24], empowering brassica by-products as a possible weapon against obesity-related metabolic dysfunctions.

This review summarizes the potential of *Brassica* by-products (focusing on broccoli) against obesity-related health problems, in a perspective of circular economy and industrial symbiosis giving an added-value to these products, once they can be a good resource of bioactive compounds [23,25].

## 2. Adipose Tissue and Adipokines in Obesity-Related Health Risks: The Role of Oxidative Stress and Antioxidant Prophylaxis

Rather than just being a deposit for triglycerides (TG), the white adipose tissue is also an endocrine organ, secreting adipokines or adipocytokines [26,27,28,29,30], with regulatory and inflammatory functions. Amongst adipokines, the most studied are adiponectin [31], resistin [32], leptin [33], tumor necrosis factor-alpha (TNF-α) [34], and interleukin 6 (IL-6) [35]. These bioactive substances create reactive oxygen species (ROS) as a by-product of their activity, leading to oxidative stress and consequently to the development of oxidative stress related diseases [36]. Oxidative stress has been proposed as one of the factors behind obesity-related health complications [37].

The increase in adiposity will lead to increased production of adipokines, creating a chronic pro-inflammatory environment in the adipocytes, characteristic of obese patients [38,39,40]. The exception is adiponectin which is inhibited by the increased secretion of IL-6 and TNF-α [41]. This inflammatory state in the adipose tissue inhibits the expression and activity of antioxidant enzymes, like catalase, superoxide dismutase and glutathione peroxidase, usually found in high quantities in the adipocytes [40]. A pro-inflammatory environment also increases the expression of NADPH oxidase, an enzyme that generates ROS as an immune response. Thus, excess adipocytes will secret excess adipokines (except adiponectin), which will induce ROS formation, increasing oxidative stress and related injury [40].

Excess of FFA in circulation, due to a high-fat diet, will increase peroxisomal and mitochondrial fatty acid metabolism, enhancing the β-oxidation in hepatocytes, and consequently, causing an electron overflow in the mitochondrial electron transfer chain [42,43], enhancing ROS formation. Furthermore, the accumulation of FFA and TG, in the form of fat droplets in the cytoplasm of hepatocytes, increases the peroxisomal pathway of fat oxidation, potentiating ROS formation that may disturb peroxisomal redox homeostasis and contribute to disease development [44].

Cancer has also been shown to have a higher incidence in obese patients. Breast, pancreatic, liver, and colorectal cancers are the most common types of cancer that have been found in obese patients. Leptin, adiponectin, and inflammation play a significant role in the mechanisms that connect obesity and cancer. In fact, in obese animal models suffering from nonalcoholic steatohepatitis (NASH), the lack of adiponectin increased hepatic tumor formation and oxidative stress generation, linking oxidative stress to cancer [45]. Further, ROS also leads to increased rates of mutations and increases susceptibility to mutagenic agents [46], which will lead to DNA modifications or damage during the early stages of carcinogenesis. ROS also play a role in tumor proliferation, promoting invasion or metastasis of cancer cells [47]. They facilitate the stabilization of hypoxia-inducible factor 1, a transcription factor of vascular endothelial growth, leading to the tumor’s angiogenesis [47].

Antioxidants are compounds that can prevent the oxidation of the biomolecules, scavenging the free radicals produced during the redox reactions. The antioxidant properties of *Brassica* are due to the presence of antioxidant phytochemicals, mainly polyphenols and ascorbic acid [48], and indirect antioxidants such as sulforaphane, an isothiocyanate derived from the hydrolysis of the glucoraphanin glucosinolate [49]. Sulforaphane, one of the bioactive compounds derived from *Brassica*, can decrease ROS generated by the augmented β-oxidation fluxes, improving mitochondrial function, maintaining its transmembrane potential and respiratory capacity, and inhibiting FFA induced mitochondrial swelling by indirectly up-regulating the transcription of antioxidant enzymes [49,50,51]. Quercetin can also help in this context since one of its possible action mechanisms is the same as sulforaphane: the activation of the Nrf2-mediated pathway [52]. Anticancer properties have also been observed in quercetin [53]. It has been suggested that quercetin induces apoptosis by directly activating the caspase cascade through the mitochondrial pathway in breast cancer MCF-7 cells [54]. Some studies also show that when conjugated with anticancer drugs such as doxorubicin, quercetin increases their cytotoxicity by interacting with ATP-Binding Cassette (ABC) transporters and allowing the drug to fight through multidrug resistance in tumor cells that are drug-resistant [55].

## 3. The Bioactive Compounds in Brassicaceae

In several epidemiological studies, the consumption of plant-based foods, a characteristic of the Mediterranean diet, has been associated with a lower risk of cardiovascular diseases and mortality [56]. In the last years, several studies, *in vitro* and *in vivo*, have been focused on the bioactive compounds of Brassicaceae and their potential to mitigate chronic diseases [48,57,58]. Dietary phytochemicals might be employed as anti-obesity agents once they suppress adipose tissue growth, inhibit differentiation of preadipocytes, stimulate lipolysis and induce apoptosis of existing adipocytes, thereby reducing adipose tissue mass [59,60,61]. As a health-promoting food, broccoli (*Brassica oleracea* var. *Italica*) has acquired considerable relevance in the last few years. The healthy food attributes assigned to broccoli, like the prevention of chronic disorders, such as cancer and cardiovascular pathologies [25], are a consequence of its high contents of bioactive phytochemicals such as glucosinolates and isothiocyanates, phenolic compounds (chlorogenic and sinapic acid derivatives and flavonoids) and nutrients (including vitamins C, E, A, K, and essential minerals: N, K, Ca, Fe, and others).

Glucosinolates are thioglucosides [62], secondary metabolites that have in their basic structure a β-thioglucose group, a sulfonated oxime group, and a variable aglycone side chain derived from an α-amino acid [63]. In broccoli, they coexist with an enzyme called myrosinase, which initiates rapid hydrolysis when in contact with water during the plant’s cutting, harvesting, or chewing. The products of this breakdown are thiocyanates, nitriles, goitrin, epithionitriles, and isothiocyanates [63,64]. This last group is well known to intervene in many steps of cancer development, more specifically in the modulation of enzymes that act in phase II detoxification, acting as an antioxidant in either a direct or indirect capacity [65]. The most studied of the isothiocyanates generated is a bioactive compound called sulforaphane [66]. This compound increases nuclear factor erythroid-2-related factor 2 (Nrf2) binding to DNA, more specifically to antioxidant response elements (ARE), responsible for the transcriptional regulation of antioxidant enzymes such as catalase, glutathione S-transferase, glutathione-peroxidase, peroxiredoxins, and hemeoxygenase [67]. A peculiarity of sulforaphane is that its antioxidant activity modulates lipid metabolism in hepatocytes, reducing lipid levels in cases of excessive accumulation, up-regulating mitochondrial gene expression, function, and mitochondrial biogenesis [51]. These properties make sulforaphane a prime candidate use against a preoccupying condition rampaging through mainly developed countries: obesity and its associated complications [24].

Phenolic compounds are one of the most relevant families of phytochemicals endowed with health benefits. They are considered efficient free radical scavengers since they can chelate redox-active metal ions, inhibit LDL cholesterol oxidation, and neutralize other processes involving ROS [68]. Moreover, dietary polyphenols may suppress adipose tissue growth, modulating antiangiogenic activity and adipocyte metabolism [59]. Further, polyphenols’ benefits are related to infections, cancer, and neurodegenerative processes [69,70,71]. One prominent polyphenol in broccoli is quercetin, whose antioxidant activity was already described in Section 2 [72].

## 4. By-Products: Circular Economy and Industrial Symbiosis View

The Food and Agriculture Organization of the United Nations (FAO) has estimated that about a third of the food produced for human consumption is either lost or wasted throughout the food supply chain, from the agricultural stage to the household consumption stage, representing a wastage of about 1.3 billion tons [73]. Furthermore, it represents wasted resources and more extensive production of greenhouse gas (GHG) emissions. Not taking into account the ecological footprint associated with the use of large expanses of farmland, there is an estimated emission of 3.3 Gtonnes of CO_2_ equivalent, a heavy carbon footprint associated with food produced and not eaten [74]. From the production side, these emissions are increased concerning wastage in two ways: on one side, farmers need to produce more of a particular product to achieve a certain amount of profit, thus increasing the emissions, and on the other side, the disposal of such a massive amount of organic matter usually involves processes such as incineration, causing further emission of greenhouse gases [73,75]. In addition, noteworthy are the GHG emissions generated in vain when a certain number of agricultural products is lost (emission per kg of product). However, to be quantified are the considerable costs to society that come from the loss of land, water, and biodiversity and the negative impact of climate change [74]. The term food loss is used when the decrease in food mass in the supply chain occurs before retail and consumption (production, post-harvest, and processing), at which point the term food waste is employed [76]. Both terms are used in the context of edible food for human consumption. Food wastage refers to both food loss and waste.

Food loss accounts for most of the wastage worldwide. When we look at the different groups of commodities, vegetables and fruits are the second most-produced category for human consumption, behind cereals. However, cereal has a lower wastage rate, and a much smaller portion of this wastage is due to agriculture. In industrialized Asia, Europe, and South and Southeast Asia, the wastage of vegetables constitutes a high carbon footprint, mainly due to large wastage volumes [73,74]. Thus, the re-use of food waste would positively impact the ecologic footprint.

In industrialized countries, production often exceeds demand due to the need to deliver agreed quantities of food, even in adverse conditions such as unpredictable bad weather or pest attacks. On the other hand, the standard set by supermarkets and other retailers for appealing fresh products (e.g., appearance, shape, color) means that large portions of the crop, which would otherwise be suitable for human consumption, never leave the farms [73,74]. Most of the food loss on the agricultural side is due to this grading system. Some surplus crops are sold to be processed or to become animal feed; this, however, is often not financially profitable [73]. The prevailing attitude in these industrialized countries is that disposal is cheaper than using or re-using [73]. For example, most wastage in the fruit and vegetable group is dumped in landfills or rivers, creating environmental hazards [75].

Amongst the waste generated in the agricultural stage, by-products are organic wastes generated that were never intended for human consumption. These by-products include leaves and stalks, harvesting, and processing waste [77]. By-products from vegetables and fruits represent 25% and 30% of the total organic mass-produced [78]. Nevertheless, non-edible materials produced by processing fruits and vegetables such as peels, seeds, and stalks, also rich in bioactive compounds, can be used as a source of phytochemicals and antioxidants [79]. The use of these by-products appears to be a clever way to mitigate the environmental problems and provide a new avenue of profit and economic rentability for companies and farmers. The use of by-products can also help bring down the price of healthy food, increasing low-income households’ access to it [73].

## 5. Broccoli By-Products Composition

Out of all the different parts of the broccoli plant, leaves and stems are the most studied. Together with the roots, they compose the non-edible fraction of the plant and therefore represent the most significant interest in valorization. However, as mentioned in Section 4, some edible parts are also thrown out, and their value beyond consumption should not be overshadowed. Characterization of these by-products is an essential step to unlock their biomedical potential. Table 1 shows the main compounds present in broccoli leaves and stems. Several compounds can vary in concentration depending on the part of the plant being studied [80], type of culture [81], the season of harvest [80], and several other factors [82]. For example, glucosinolates exist in greater quantity in the stems than in the leaves [80,81,83]. However, glucosinolates can also exist in greater amounts in the leaves than in the stems [25], because glucosinolate levels may also vary depending on the age of the plant, the time of harvest, or the method being used for quantification. Less mature parts of the plant are wealthiest in glucosinolates [82]. Overall, a higher concentration of glucosinolates is found in broccoli plants cultivated with organic procedures rather than conventional procedures [81], and under spring conditions over fall conditions [80]. Myrosinase hydrolyzes glucosinolates to isothiocyanates and other metabolites, some of them with toxic properties. One factor that affects the concentration of glucosinolate metabolites in the plant is the activity of the enzyme myrosinase. The activity of myrosinase also varies according to the plant part [80].

On the other hand, in *Brassica oleracea* plants, the highest content of polyphenols (such as flavonoids) is found in the leaves [20]. Sulfur-based nutrition increases their phenolic content, while nitrogen-based nutrition causes its decrease [20]. Like glucosinolates, a higher concentration of phenolic content is detected in plants grown in organic cultures when compared to plants grown with conventional methods [20]. Using by-products from organic cultures could have more advantages beyond compound concentration. Conventional methods employ the use of chemical pesticides and other products not used in organic cultures. Although no studies have been performed specifically on *Brassica oleracea* waste, it was observed in other species that 30% of the parts used for consumption contained pesticides when conventional culture methods were used [84]. No pesticide was found in concentrations higher than the acceptable daily intake; however, the risk posed by cumulative exposure to some of the compounds found in these pesticides (such as sodium channel modulator compounds) could become a concern for consumer health [84]. *Brassica oleracea* is also rich in essential minerals distributed all over the plant. Iron (Fe), zinc (Zn), and phosphorus (P) were found in more significant quantities in the inflorescence, Manganese (Mn) and calcium (Ca) in the leaves, and sodium (Na) in the stalk [25].

The characterization of the plant’s by-products and how the various components change according to the different conditions the plants are exposed to is vital to determining their future use and applications. The characterization should also include an assessment of the levels of pesticides present on the by-products for consumer health concerns. For example, the most commonly found pesticide in *Brassica oleracea*, chlorpyrifos [84], inhibits cholinesterase [85] and may cause excessive stimulation of the parasympathetic nervous system, leading to symptoms such as hypermotility, hypersecretion, miosis, hypotension, diarrhea, and bradycardia [86].

## 6. Added-Value of Broccoli By-Products

Many by-products of the agri-food industry may be helpful as a source of nutrients and potentially functional ingredients, allowing obtaining added-value products. In the case of broccoli, the marketable florets (flower heads) represent only a minor part of the total above-ground plant biomass (<25% of total yield), generating a vast amount of crop remains [25,87]. Moreover, the abnormally high temperatures in the winter and spring seasons may induce premature flowering, resulting in the total loss of the marketable yield (heads) and converting all the biomass into an unprofitable by-product [32], constituting a significant amount of waste, with a negative effect on the agricultural environment. Adding value to broccoli by-products is very important, as recycling all this agro-waste to obtain bioactive ingredients for the industry can improve profitability and reduce costs and environmental problems. Furthermore, the potential use of vegetable by-products as a source of bioactive compounds is drawing the attention of the scientific community [88]. Broccoli by-products have similar chemical composition to broccoli florets, being also rich sources of glucosinolates, polyphenols, dietary fiber, proteins, and vitamins [25,83,89]. However, broccoli by-products are currently limited to flour and fiber [89] and glucosinolate standard extraction or characterization [90] and are therefore underutilized. Recently, several studies have observed the usefulness of broccoli by-products in producing functional foods enriched with bioactive compounds, such as beverages, snacks, bread, or cakes [91,92,93,94]. In addition, broccoli flours can be a good source of phytochemicals as dietary supplements, as broccoli flours maintain an adequate nutritional composition and physicochemical properties [95]. Despite the above, broccoli leaves or stems are rarely added to food products, a trend that must be changed.

## 7. Conclusions

The broccoli harvest, among other *Brassica* crops, generates large amounts of organic material filled with valuable compounds that are thrown away daily. Scientific evidence shows that these bioactive compounds, with great antioxidant potential, can play an essential role in preventing or mitigating chronic diseases such as obesity and associated health disorders. In addition, broccoli by-products are suitable for functional foods production. This review intends to show how beneficial the wasted by-products of broccoli can be, promoting a significant effort to repurpose them and creating health, economic, social, and environmental value. Of particular interest, the normalization of cellular metabolism may be an essential target for the active compounds in *Brassica* by-products, namely those harboring anti-inflammatory and antioxidant activity, thus reverting or minimizing some of the complications associated with obesity in different organs.

## Figures and Tables

**Table 1 nutrients-13-04194-t001:** Major compounds present in broccoli by-products (leaves and stems).

Compounds			Reference
Glucosinolates	Aliphatic Glucosinolates	Glucoraphanin, Glucoiberin, Progoitrin, Gluconapin, Glucoerucin, Sinigrin	[25,83]
	Aromatic Glucosinolates	Gluconasturtiin	
	Indolyl Glucosinolates	Glucobrassicin, 4-Hydroxyglucobrassicin, 4-Metoxyglucobrassicin, Neoglucobrassicin	
Phenolic compounds	Total Phenols		[25,83]
	Hydroxycinnamic acid derivatives	Chlorogenic acid derivatives, Sinapic acid derivatives	
	Flavonoids		
Vitamins	Vitamin C Vitamin K1 Vitamin E		[25,83]
Mineral nutrients	C, N, P, S, Na, K, Ca, Mg, Fe, Mn, Zn, Cu		[25,83]
Photosynthetic pigments	Chlorophylls	Chlorophyll *a* Chlorophyll *b*	[25]
	Carotenoids	β-carotene, Violaxanthin, Neoxanthin, Lutein	

## Data Availability

Not applicable.

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
