# Peer review of "Three in One: The Potential of Brassica By-Products against Economic Waste, Environmental Hazard, and Metabolic Disruption in Obesity"

_nutrients, 2021, doi:10.3390/nu13124194_

Round 1

Reviewer 1 Report

This study is a good review that accurately analyzes the issue of food loss, which is particularly important in recent years, from a public health perspective. On the other hand, there are some issues that need to be fixed.

1) Possible reasons for partially removing broccoli include the use of pesticides and measures against stains. In particular, the effects of pesticides are not mentioned much in this review, so please add them.

2) While broccoli is widely used in hospital foods, it used as some patient reasonable care not only obesity. Please explain in more detail why this review targeted for only obesity. Alternatively, show on its effectiveness for outside of obesity.

3) About broccoli, please mention the differences between countries and production areas. For example, in the case of rice, the varieties vary greatly depending on the Asian countries where they are consumed in large quantities, so the effects on obesity may also differ.

4) Please add any possible disadvantages in reducing the amount of broccoli discarded.

Author Response

The authors are very grateful to the reviewer for the input given to this manuscript. Improvements in the quality of the manuscript have been made based on the reviewer’s helpful and interesting comments. Below are our answers to the reviewer’s comments, provided point-by-point. 

Comment1:  Possible reasons for partially removing broccoli include the use of pesticides and measures against stains. In particular, the effects of pesticides are not mentioned much in this review, so please add them. 

Our answer:  This is an interesting point that deserves to be discussed in our manuscript. Concerns about the effects of pesticides have been addressed in our review (line 313-321 and line 327-332).

Comment2: While broccoli is widely used in hospital foods, it used as some patient reasonable care not only obesity. Please explain in more detail why this review targeted for only obesity . Alternatively, show on its effectiveness for outside of obesity.

Our answer: We thank the reviewer for the comment. As we reported in the introduction of our manuscript,  obesity has reached epidemic proportions worldwide, and it has risen at an alarming rate and becoming a significant public health concern with a vast social cost. Obesity leads to metabolic disorders, such as type 2 diabetes, non-alcoholic fatty liver disease, hypertension, coronary heart disease, several cancers, and psychological problems. Since our laboratory is interested in this topic, we focused our review on it.

Comment 3: About broccoli, please mention the differences between countries and production areas. For example, in the case of rice, the varieties vary greatly depending on the Asian countries where they are consumed in large quantities, so the effects on obesity may also differ.

Our answer: According to the Food and Agriculture Organization of the United Nations (fao.org) website, the production of cauliflowers and broccoli has been growing in the last two decades. The major production share is from Asia (77%), following Europe (11.3%) and the Americas (9.4%). The top 3 producers are effectively Chine, India, and the United States of America. Being the major producers does not necessarily mean being the major consumers. In the authors’ opinion, including that information in the manuscript may introduce entropy and misinformation, so we decided not to include it. However, we have listed in the article are a series of factors that may impact the composition of broccoli. Further, it is described how some of these factors affect specific compound concentrations in the plant.          

Comment 4: Please add any possible disadvantages in reducing the amount of broccoli discarded.

Our answer:  The authors’ goal with this manuscript is to draw readers’ attention to the possibility of reusing food waste. By finding strategies for reusing byproducts, that may even have therapeutic effects, we contribute to more sustainable agriculture and promote good health and well-being, goals included in the agenda 2030.  Thus, the authors fail to see any disadvantages in reducing the amount of broccoli discarded.

Reviewer 2 Report

The title of this is a bit convoluted, and the theme of the paper is excellent and will be a good contribution to the literature.  However, it does feel like a mash-up from someone’s thesis that is not put together well.  It is overbalanced with a ponderous pseudo-academic review of each of the three subjects, followed by short descriptions of how Brassica waste could help.  Again, a noble goal that has great place in our societies right now, but it whipsaws back and forth too much between deep review and futuristic projections, with no [NEW] specific suggestions for precisely what the authors are proposing be made from this 75% of plant biomass that is now wasted.  In other words you lay out the problem nicely, and review a bunch of literature ancillary to some of the potential solutions to the problem, but don’t really get your teeth into what you propose doing about the problem other than “utilizing the rest of the plant for health- and environment-friendly solutions” (my words, not yours).  So my suggestion is to radically condense and shorten (by about ½), and to pay more attention to what you are actually recommending.

Neither Fig 1 nor Table 1 necessary or helpful.  It would be better to have a more industrial process flow diagram indicating the multiplicity of suggested pathways in which this “waste” plant material might be utilized, and suggesting some tonnage, etc.

L 105 – “Hight” – please check useage and spelling

L100, 111, 112 – each use IL-6 and spell out – one has italic – get with a protocol as per journal (e.g. first use of abbreviation  -- you define, and then use abbreviation thereafter

L100, 118 – same for TNF-a

Please vet document for similar inconsistencies of abbrev.

L 146 – the symbol for reversible rxn looks odd as placed with a box around it

L 170 – glucosinolates (themselves) are not antioxidants; sulforaphane, is an indirect antioxidant which up-regulates antioxidant enzymes.  This point should be made and properly referenced (e.g. Joko et al 2017 -- https://doi.org/10.1016/j.jff.2017.05.039).

Section 2. – You spend almost a page and a half doing a review that has nothing (directly) to do with the Brassica vegetables (the theme of your paper) then have a single paragraph a the end about Brassica phytochemicals.  This would be better dealt with by references to full-on reviews of the subject

L 195 – What on earth does “nitrogen-sulfur compounds” mean?  So many phytochemicals contain both or those elements that this is a misleading statement and a red-herring, regardless of how many times it is said incorrectly in the lay press and the scientific literature.

L 211 “particularity” could perhaps be replaced with a better word here?

L 286-287 – this is confusing although I believe accurate -à are you saying that it can happen both ways depending upon the research paper you are looking at?

L 292-293 – This is inaccurate!: “Another factor that conditions the concentration of glucosinolates in the plant is the activity of the enzyme myrosinase.” Myrosinase only acts on glucosinolates after tissues are disrputed or lysed or chewed, thus the amount of glucosinolates present in the plant tissue as grown, is independent of myrosinase content and activity.

Section 6 – The second half of this section is written in a very confusing fashion and needs to be cleaned up.

Refs 5, 6, 64, 65, 94, 97, and others are just author initials and abbreviations, clearly downloaded carelessly and not checked.

Section 7 and elsewhere – “Brassica” is italicized and not – please decide which convention is proper and which 

Author Response

The authors are very grateful to the reviewer for the input given to this manuscript. Improvements in the quality of the manuscript have been made based on the reviewer’s helpful and interesting comments. Below are our answers to the reviewer’s comments, provided point-by-point. 

Comment 1: Neither Fig 1 nor Table 1 is necessary or helpful.  It would be better to have a more industrial process flow diagram indicating the multiplicity of suggested pathways in which this  “waste” plant material might be utilized, and suggesting some tonnage, etc.

Our response: The authors are grateful to the reviewer for this comment. We agree with de reviewer that figure 1 is not an added value in the context of this manuscript, and for that reason, we remove it. Regarding table 1, the authors do not have the same opinion as to the reviewer. The goal of the table is to show the compounds that have been wasted with by-products.  In our opinionw, removing the table would also remove potential from our goal. Regarding the suggested diagram, it has been included as graphical abstract.

Comment 2: L 105 – “Hight” – please check useage and spelling

Our response:  The word is now corrected.

Comment 3: L100, 111, 112 – each use IL-6 and spell out – one has italic – get with a protocol as per journal (e.g. first use of abbreviation  -- you define, and then use abbreviation thereafter).  

Our response: The abbreviations were corrected.

Comment 4: L100, 118 – same for TNF-a

Our response: The abbreviations were corrected.

Comment 5: Please vet document  for similar inconsistencies of abbrev

Our response:  The inconsistencies have been fixed.

Comment 6:  L 146 – the symbol for reversible looks odd as placed with a box around it

Our response: Thank you for the warning. It was a formatting problem. Since the equation is not fundamental, we remove it to avoid future formatting problems.

Comment 7:  L 170   – glucosinolates (themselves) are not antioxidants; sulforaphane, is an indirect antioxidant which up-regulates antioxidant enzymes.  This point should be made and properly referenced (e.g. Joko et al 2017 -- https://doi.org/10.1016/j.jff.2017.05.039).

Our response:   Thank you for the comment. We agree with the reviewer, and the sentence has been rephrased.

Comment 8:  Section 2. –   You spend almost a page and a half doing a review that has nothing (directly) to do with the Brassica vegetables (the theme of your paper) then have a single paragraph a the end about Brassica phytochemicals.  This would be better dealt with by references to full-on reviews of the subject.

Our response: we shortened section 2 by removing non-relevant information to the manuscript’s topic, as suggested by the reviewer.

Comment 8: L 195 – What on earth does “nitrogen-sulfur compounds”  mean?  So many phytochemicals contain both or those elements that this is a misleading statement and a red-herring, regardless of how many times it is said incorrectly in the lay press and the scientific literature.

Our response: By “nitrogen-sulfur compounds,” we mean compounds like isothiocyanate or glucosinolate and its derivates.    However, to avoid misunderstandings, the statement “nitrogen-sulfur compounds” has been removed.

Comment 9: L 211 “particularity”  could perhaps be replaced with a better word here?

Our response: The word has been replaced.

Comment 10: L 286-287  – this is confusing although I believe accurate -à are you saying that it can happen both ways depending upon the research paper you are looking at?

Our response:  The sentence has been rephrased. “For example, glucosinolates exist in greater quantity in the stems than in the leaves. However, glucosinolates can also exist in greater amounts in the leaves than in the stems, because glucosinolate levels may also vary depending on the age of the plant, the time of harvest, or the method being used for quantification”

Comment 11: L 292-293  – This is inaccurate!: “Another factor that conditions the concentration of glucosinolates in the plant is the activity of the enzyme myrosinase.” Myrosinase only acts on glucosinolates after tissues are disrputed or lysed or chewed, thus the amount of glucosinolates present in the plant tissue as grown, is independent of myrosinase content and activity.

Our response:  We agree with the reviewer, this is inaccurate. The sentence has been rephrased. “Myrosinase hydrolyzes glucosinolates to isothiocyanates and other metabolites, some of which are toxic. One factor that affects the concentration of glucosinolate metabolites in the plant is the activity of the enzyme myrosinase”.

Comment 12:  Section 6 – The second half of this section is written in a very confusing fashion and needs to be cleaned up.

Our response: Section 6 has been reworded, and we hope it is more precise than before.

Comment 13: Refs 5, 6, 64, 65, 94, 97,  and others are just author initials and abbreviations, clearly downloaded carelessly and not checked.

Our response:  Refs have been corrected.

Comment 14: Section 7 and elsewhere – “Brassica” is italicized and not – please decide which convention is proper and which

Our response: The inconsistencies have been fixed.